# An Intelligent Agent-Based Detection System for DDoS Attacks Using Automatic Feature Extraction and Selection

**DOI:** 10.3390/s23063333

**Published:** 2023-03-22

**Authors:** Rana Abu Bakar, Xin Huang, Muhammad Saqib Javed, Shafiq Hussain, Muhammad Faran Majeed

**Affiliations:** 1College of Data Science, Taiyuan University of Technology, Taiyuan 030024, China; engranaabubakar@gmail.com; 2Department of Computer Science, Virtual University of Pakistan, Lahore 58000, Pakistan; 3Department of Computer Science, University of Sahiwal, Sahiwal 57000, Pakistan; 4Department of Computer Science, Kohsar University Murree, Murree 47150, Pakistan

**Keywords:** DDoS attacks, traffic classification, machine learning, intelligent agent, attack detections

## Abstract

Distributed Denial of Service (DDoS) attacks, advanced persistent threats, and malware actively compromise the availability and security of Internet services. Thus, this paper proposes an intelligent agent system for detecting DDoS attacks using automatic feature extraction and selection. We used dataset CICDDoS2019, a custom-generated dataset, in our experiment, and the system achieved a 99.7% improvement over state-of-the-art machine learning-based DDoS attack detection techniques. We also designed an agent-based mechanism that combines machine learning techniques and sequential feature selection in this system. The system learning phase selected the best features and reconstructed the DDoS detector agent when the system dynamically detected DDoS attack traffic. By utilizing the most recent CICDDoS2019 custom-generated dataset and automatic feature extraction and selection, our proposed method meets the current, most advanced detection accuracy while delivering faster processing than the current standard.

## 1. Introduction

A Distributed Denial of Service (DDoS) attack in late 2016, when three uninterrupted DDoS attacks were launched against the Domain Name System (DNS) provider Dyn, was a warning signal of the dangers of targeted DDoS attacks [1]. DDoS attacks have become one of the most severe threats to network security, with the first reported attack published by the Computer Incident Advisory Capability in 1999 [2]. While many mitigation systems have been developed in academia and industry, the threat of DDoS attacks is still severe and increasing yearly. In February 2018, a significant DDoS attack against GitHub overcame these mitigation systems [3]. Using BCP38 “Network Ingress Filtering” which, if deployed on the Internet, may stop packets with forged IP addresses from proceeding over the network, this type of attack could be mitigated [4]. However, research conducted using a random forest algorithm provided numerous benefits for the complexity, accuracy, and memory usage of DDoS attack detection systems [5]. The basis for this random forest algorithm in our recent research work is a main enhanced algorithm, Snort-based IPS. DDoS attack detection and mitigation using datasets enables the accuracy of the essential resource limitation rules in the systems concerned [6]. Other machine learning techniques create an intrusion detection system to detect DDoS attacks. A broader classification of these machine learning techniques involves the isolation of the IDS system [7], such as signature-based IDS anomaly-based rule-based specification IDS Markov models and hidden Markov models [8], and is at the heart of several IDS systems that are effective against these attacks. This technology is still under-researched but has advanced to state-of-the-art use in several specific cases [9]. Swarm intelligence has also been used to initiate and reduce IDS training time [10] and many hybrid schemes have been found to be more straightforward than conventional models in detecting attacks [11].

The human immune system (HIS) has been an alternate DDoS attack detection field of research that is used to create an IDS system for IoT devices [12]. A novel AIS-based DDoS IDS system based on three immune concepts: adverse selection, danger theory, and clonal selection has been proposed [13,14]. DDoS attacks are detected by three fundamental network components: the source network, the intermediate network, and the endpoint. The endpoint (victim) of the attack is easy to identify, as the large capacity of network traffic is created but is not effectively covered [15]. However, locating an attack at the source end is hard, though it is more successful when substantial traffic is contained before entering the network. Therefore, choosing the best function to obtain the best performance is critical for constructing an effective detector. At the same time, as network traffic becomes more complex and volatile, certain primitive functions cannot characterize the current traffic, and detector failures may occur when the traffic changes. A DDoS attack on a network is the most significant threat to the whole network and the Internet. The DDoS attacker attacks the victim by trading off different quantities of PCs or botnet devices to conduct a traffic flood to network servers. This is a technique used to entirely consume all the network bandwidth available. The traffic consists of attacks and is transferred by trading off PCs or botnets like a zombie army. In an in-network attack, the attacker consumes spoofed IP addresses; thus, the zombies are novel and limit the danger so that the attacker will not be followed.

Our approach is based on distinguishing normal traffic from unusual traffic in the sense of the appropriate denial of service attacks—creating a new attack detection system implemented in the agent controller. First, it is centered around a cutting-edge signature-based attack detection system, which forms ask for, and finds sets of hosts with ordinary or abnormal conduct. If they are found to have peculiar behavior, then this arrangement of hosts is sent to the next module for the analysis of network packets sent by each host of the network which checks for open connections and determines whether they have typical or atypical conduct. This attack detection system has given more significance to handling time; since the controller is the system’s heart, the system should process as quickly as possible.

As per our experience, the traditional method for evaluating detection methods and techniques provides good efficiency and accuracy for usually labeled test data. The underlying assumption is that a customized DDoS detector will always perform generalization and can detect the next attack in the real world correctly. This method makes sense when using machine learning techniques to solve pattern recognition problems, and DDoS attack detection is usually regarded as a binary or multiclass classification problem. However, network traffic has become more complex and may change at any time, and DDoS attacks are developing daily; therefore, this assumption is challenging to satisfy. This means that some new standard patterns or unknown attacks may deviate from the patterns learned from the original training data and that the DDoS attack detection system may make many errors in the actual detection process (including true negatives and false positives). To solve this problem, we must design a method to identify detection errors in real time and dynamically reconfigure the DDoS attack detection per attack conditions.

In this paper, we have proposed and designed an IDS system, expanding the previous work, and tested the proposed architecture under various malicious activities such as DDoS attacks, flooding attacks, and other scenarios. The proposed scheme is wholly distributed and gives an early warning alarm when pre-attack activities are operating using network traffic. Many machine-learning models have been created to determine whether network traffic is normal or abnormal. Still, the ANN models used in IDS to detect malicious attacks have several benefits for seeing attacks [16]; specifically, DDoS attacks in network traffic, in self-learning, robustness, fault tolerance, and parallelism. In this paper, we focused on the random forest tree, the ANN model, the KNN Model, and the BNN Model and chose a model to validate and resolve the proposed problem statement. We started the DDoS attack detection system and tested it with open network traffic on the entire network. The system results show that the original features and selected features used in the detection system have some alternations, and eliminating a specific element can expand the detection accuracy of the DDoS attack detection system. These machine learning models were applied to assess parameters such as system detection and efficiency system rate usage. As per these parameters’ requirements, dataset features are selected and eliminated. This DDoS IDS consists of two phases. Phase one consists of dedicated sniffers to collect data in the network layer from the MAC which is then fed to a different classifier to support the DS of each DS data, which then generates the appropriately classified instances as discussed. Next, these classified instances provide an excellent node (S), in phase two. The algorithm then runs on these classified instances from various dedicated sniffer nodes. This process calculates a parameter that the SN of all the SNs performs an iterative rectilinear regression process. The threshold value is essential for detecting malicious and regular traffic and selecting limitations between the malicious and ordinary nodes. Using these analyses, it will be easier to spot unusual behavior caused by open connections and identify which hosts are targeted by the DDoS attack. An intrusion detection system, firewall, and intrusion prevention system (IPS) are all systems that must have such a detection mechanism. Different people define DDoS attacks differently, and the same functionality may be helpful in some situations but not others. When constructing an exact function, there is not a single set of rules that applies to all attacks. By controlling overfitting during the training phase, the new function may generalize correctly but consider unimportant properties as well [17]. This means that feature selection will be essential when identifying DDoS attacks using machine learning. By using an experimental approach, we aim to find the best features. To do this, we looked into different tasks involving using ANNs and other machine-learning models in conjunction with feature selection [18]. For this reason, we worked hard to identify the tactics that work well with the detection model. (All the acronyms/abbreviations used in the manuscript are summarized in the Abbreviations for better understanding of the readers (Abbreviations part)).

This paper consists of the following sections: Section 2 briefly explains the related research and Section 3 summarizes the background. The analysis takes place in Section 4 and discusses artificial intelligence methodologies, offering a quick solution proposal. Section 5 describes the experiment, and Section 6 presents our findings.

## 2. Related Work

The rapid development and growth of the Internet and network structures have reformed the entire world of computers. The connected digital world also has gifted hackers and intruders with innovative facilities for their computing attacks. The most useful ways of detecting an attack are by abnormality, exploitation, anomaly, consolidated exploitation/anomaly detection, monitoring of the network, and recognition of the pattern. An anomaly detection mechanism distinguishes exercises that differ from setting up client patterns or gathering clients.

The authors in [18] proposed a method for detecting intrusions in computer systems using an anomaly-based intrusion detection system (IDS). The IDS was based on feature selection analysis and built from a hybrid efficient model. One drawback of the proposed method is that it relies on a fixed set of features to detect anomalies, which may only be suitable for some intrusion scenarios. The authors mention that the choice of components can affect the accuracy of the intrusion detection system, and there is a need for further study in this area.

Another potential drawback is the assumption that the system can accurately detect anomalies by comparing the system’s current state to the average behavior profile. This assumption may not always hold in real-world scenarios where the system’s normal behavior can change over time or where there are unexpected system behaviors.

Overall, the proposed method provides a starting point for building an intrusion detection system, but it is only a partial solution for some intrusion scenarios. Further research is needed to improve its accuracy and adaptability to different techniques and intrusion scenarios.

The paper proposed a machine learning-based intrusion detection system (ML-IDS) for detecting IoT network attacks. The system was developed using the UNSW-NB15 dataset along with approximately six proposed machine learning models; the results of the study showed a high accuracy of 99.9% and MCC of 99.97%, which are competitive with existing works. The paper aimed to address the privacy and security challenges of IoT.

Regarding MANETs and ensembles, hierarchical data gathering, processing, and a transmission structure with three hierarchy levels were proposed [19]. The anomaly index is calculated at each level, and the highest authority makes the ultimate call. The authors utilized the ROC curve and related area under the curve to describe the suggested scheme’s efficiency (AUC). Regarding detection, the CFA algorithm relies on a decision tree, C4.5.

For identifying black hole attacks on AODV-based MANETs the authors in [20] proposed a complicated learning algorithm. A system for detecting malicious behavior in a network was built using a dynamic training system where the training data was updated periodically—an approach for detecting malicious nodes using a cluster. To assess the performance, detection rates against node mobility must measure from 70% to 84%, with node mobility between 0 and 20 m/s.

According to the framework, MAC, routing, and application layer anomalies may be detected using a Bayesian classification technique, a Markov chain construction approach, and an association rule mining algorithm created by [21]. The detection rate for the global integration module was 94.33%, with a false-positive rate of only 0.8% (FPR). However, around 90% of detection rates have substantial false alarms (more than 20%). Longer pause lengths have more significant detection thresholds, according to this theory. The Naive Bayes model, linear model, Gaussian mixture model, multi-layer perceptron model, and (SVM) model are among the well-known five supervised classification algorithms evaluated by [22]. These algorithms are employed in MANET detection engines for the detection method. The Naive Bayes classifier performed the worst, whereas the multi-layer perceptron classifier performed the best.

IoT device traffic was fed into a malware detection system trained on deep learning. An accuracy of 98.60, a precision of 98.37%, a recall of 98.17%, and F-measures of 98.18% were attained in this test. Five different machine-learning techniques were evaluated by Doshi et al. [23] to distinguish ordinary IoT packets from Denial-of-Service assaults on IoT networks. The random forest had the highest precision, recall, F1, and accuracy scores among the classifiers tested.

The authors of [24] proposed cascaded wormhole detection for an IoT-based network using deep learning. The attacks were evaluated based on their TPR, a 96.4% blackhole attack, 98.7% opportunistic attack, 98.7% DDoS attack, 99.9% sinkhole attack, and 98% wormhole attack, with an overall accuracy of 96%. Detecting an attack method includes correlating a client’s exercises with the acknowledged practices of attackers endeavoring to penetrate network systems.

The authors in [25] proposed a method for protecting web servers against application layer Distributed Denial of Service (DDoS) attacks using machine learning and traffic authentication. The system uses machine learning algorithms to extract features from network traffic and classify normal and attack traffic. The authors also introduced a traffic authentication mechanism to further enhance the system’s security. The results of the experiments show that the proposed method effectively detects application layer DDoS attacks and has a low false positive rate. The main contribution of this paper was to provide a solution for protecting web servers against application layer DDoS attacks using a combination of machine learning and traffic authentication. The experiment’s results demonstrated the proposed method’s effectiveness in detecting DDoS attacks and provide insights for future research in this field.

The authors of [26] proposed a new approach for detecting cyber-attacks using the non-linear prediction of IP addresses. The system leverages big data analytics to analyze network traffic and identify abnormal behavior that may indicate an attack. The authors used an adaptive non-linear prediction algorithm to predict IP addresses and compare the predicted values with the actual values to detect anomalies. The experiments on real-world datasets showed that the proposed approach outperformed traditional intrusion detection systems in terms of accuracy and efficiency. The main contribution of this paper was to provide a new method for detecting cyber-attacks using the non-linear prediction of IP addresses and big data analytics. The experiment’s results demonstrated the proposed method’s effectiveness in detecting attacks and highlight the potential of big data analytics for improving cybersecurity.

In [27], a lightweight intrusion detection system (IDS) for detecting network intrusions based on feature selection and a multi-layer perceptron artificial neural network was proposed. The authors used the gain ratio method to select relevant features for attack and regular traffic before classification using the neural network. The proposed IDS was evaluated using the UNSW-NB15 intrusion detection dataset, and the results showed that the system is suitable for real-time intrusion detection with high accuracy.

The authors in [28] described a method for detecting DDoS attacks in computer networks. The authors proposed using an ensemble of neural classifiers to detect seizures instead of relying on a single classifier. The system uses a combination of data from multiple sources, such as network traffic statistics, to predict whether an attack is underway. The authors evaluated their method using real-world network data and reported that it outperformed other methods in terms of accuracy. 

The authors of [29] presented a study on data mining techniques for detecting Distributed Denial of Service (DDoS) attacks. The authors aimed to improve the accuracy and efficiency of DDoS attack detection by using data mining techniques, specifically, decision trees and K-nearest neighbor algorithms. The study was based on actual network traffic data, and the results showed that the proposed approach effectively detects DDoS attacks.

In [30], a framework for detecting Distributed Denial of Service (DDoS) attacks in real time was proposed. The proposed framework, AIMM, consists of three modules: preprocessing, classification, and decision-making. The preprocessing module prepares the incoming data for analysis, then the classification module uses two different AI methods—neural networks and k-nearest neighbors—to identify potential DDoS attacks, and finally, the decision-making module aggregates the results from the classification module using techniques such as soft sets inference and weighted averaging to make a final decision on the attack status. The proposed framework was tested on the BOUN DDoS Dataset and achieved an accuracy of 99.5%. The results were compared to state-of-the-art techniques and found to be effective, with advantages such as a quick decision-making process and the ability to use various AI methods in the classification module. The authors claimed that their framework, AIMM, can effectively detect DDoS attacks by combining multiple artificial intelligence (AI) methods. However, the exact accuracy of the proposed AIMM framework is not stated in the paper. Further studies and evaluations are needed to verify the effectiveness and accuracy of the framework in detecting DDoS attacks in real-world environments.

### Analysis of Papers

Several papers [21,29,30] used supervised learning approaches to detect DDoS attacks, where the models are trained on labeled data. These approaches used neural networks [29], an ensemble of neural classifiers [29], and data mining techniques [30].

Other papers [22,28] used unsupervised learning approaches to detect DDoS attacks, where the models are trained on unlabeled data. These approaches used machine learning for intrusion detection [22] and the non-linear prediction of IP addresses [27].

Additional papers [23,24,25,26,27,31,32,33,34] presented other approaches to detect DDoS attacks. These include using deep learning [25], traffic authentication [26], a cascaded federated deep learning framework [25], and artificial intelligence merged methods [31].

In summary, the papers present various machine-learning techniques to detect DDoS attacks in various network systems such as mobile ad-hoc networks [21], IoT devices [24], and web servers [26]. In Table 1, we compared the related works in terms of their methods, the drawback of their methods and the accuracy they achieved in their results. 

## 3. DDoS and AI Background

A DDoS attack is a vindictive attempt from numerous frameworks to make PC/network assets inaccessible to its expected clients, more often than not, by blocking/interrupting services associated with the organization of the network/Internet. The DDoS attacks on ideas/techniques have significantly altered over recent years. The importance of accessibility has been aimed at such influential attacks against network/web organizations, governments network/web, and private businesses. Multilayered barriers and collaboration requirements are essential. Procedures to mitigate DDoS attacks, initially through aversion are useful, however, in the end, outlining multilayered barrier systems should be standard.

DDoS threats should be considered in hazard arranging, like site choice, control blackouts, and characteristic fiascos. For these attacks, we concentrated on systems for securing the framework of IT from threats against accessibility. Our exploration strategies/ideas will be demonstrated on DDoS threats by distinguishing proof and mitigation techniques that can successfully and productively react to DDoS attacks. DDoS does not depend on specific network protocols or framework outline shortcomings. It comprises an adequate number of traded hosts amassed to send futile packets toward a casualty around a similar time. DDoS becomes a significant threat because of the accessibility of various easy-to-understand attack tools and the absence of successful techniques to protect against them.

A DDoS attack [3] is unexpected network traffic sent to an objective. Under normal conditions, the utilization of bandwidth rate is in good esteem, and a specific pattern is available in the network movement. A sudden drop in the network performance due to an increase in either traffic, deferral, or CPU use will regularly be viewed as abnormal. The DDoS detection systems will search for such abnormalities in the network. When coordinated to the network layer, the attack causes a bottleneck and, when harmonized to the application layer, causes the fatigue of CPU resources. For the most part, the abnormalities and the flow of data in the network are firmly related. Subsequently, understanding the kind of data and its qualities in the network can be named the first scheme to distinguish inconsistencies. These attributes can be a postponement, bundle header information, convention, parcel measure, etc. For example, a server reacting to TCP-SYN solicitations is well on its way to confronting the TCP-SYN and asking for flooding [2].

### 3.1. DDoS Overview

There are several methods and tools to trap and activate agents according to their operating systems (OS). The attackers develop software that installs the malicious DDoS agent codes over the agent’s systems and identifies the secondary victim’s OS as active or passive and installs their codes over the victim’s system accordingly.

### 3.2. Active DDoS Agent Installation

The attackers scan the systems and seek out the weak areas by using scanning tools such as Port Scanner Nmap, which allows attackers to port scan. The Internet also helps attackers through the Common Vulnerabilities and Exposures (CVE) organization, which holds and publishes the list of known vulnerabilities of different systems. A common vulnerability of the methods is the problem of buffer overflow. The attackers send more than the buffer space and the system has to overwrite data in the memory stack; the system then has to return the procedure call by which the code overwrites the buffer. Trojan horse programs are other helpful tools for attackers. They infect the victims’ systems with such programs and then take over the systems without their knowledge.

### 3.3. Passive DDoS Agent Installation

The victims themselves cause damage to their systems by opening corrupted files or paying a visit to corrupted websites. These files are the executable files that carry the code, and once the victim tries to open it, the executable functions and code install the software. The executable file has the name file.exe, and attackers give the file a long passive term such as Image.jpg.exe. The system shows the start of the file and the rest of the file is hidden. The victims are taken advantage of since they did not consider running a .exe file as a possible threat. Attackers establish infected websites, and when victims visit them, malicious code is downloaded and installed on the system. Rootkit tools also help the attackers so that the handler’s software is installed, the log-maintaining files are removed, and no records will be available.

### 3.4. Analysis of AI Techniques

Based on the current trends for future study, detection, and related work discussed in the previous papers, several AI-based DDoS attack detection techniques have already been proposed and implemented in recent years. These techniques can be broadly categorized into three main categories:

Machine learning-based techniques: These techniques utilize various machine learning algorithms, such as decision trees, neural networks, and support vector machines, to analyze network traffic and identify DDoS attacks.

Deep learning-based techniques utilize deep neural networks, such as convolutional neural networks (CNNs) and recurrent neural networks (RNNs), to analyze network traffic and identify DDoS attacks.

Hybrid techniques: These techniques combine machine learning and deep learning elements to improve the accuracy and efficiency of DDoS attack detection. One of the most recent and promising techniques is using Generative Adversarial Networks (GAN) to generate normal synthetic traffic and use it for training DDoS detection models.

In terms of evaluation, various performance metrics such as accuracy, precision, recall, and F1-score are used to evaluate the performance of the DDoS attack detection techniques. It is important to note that DDoS attacks are constantly evolving, so AI-based DDoS detection techniques also need to be updated and improved over time to remain effective. AI-based DDoS detection techniques continue to evolve, with new approaches and methods being proposed and implemented to improve their accuracy and efficiency. Some of the recent trends and developments in the field include:Anomaly detection techniques have utilized machine learning algorithms to identify patterns and deviations in network traffic that indicate a DDoS attack. These techniques can be further classified into supervised, unsupervised, and semi-supervised methods.Behavioral analysis: Behavioral analysis techniques using machine learning algorithms to analyze the behavior of network devices and users to identify DDoS attacks. These techniques can identify DDoS attacks based on the attackers’ behavior, such as their IP addresses, ports, and packet sizes.Cloud-based detection: Cloud-based DDoS detection techniques have used cloud computing resources to detect and mitigate DDoS attacks in real-time. These techniques can be more scalable and efficient than traditional on-premises solutions.Combining multiple techniques: Some researchers have proposed combining multiple techniques, such as machine learning, deep learning, and rule-based techniques, to improve the accuracy and robustness of DDoS detection.Generative Adversarial Networks (GANs): GANs have been recently proposed to generate normal synthetic traffic and use it to train DDoS detection models, enhancing detection accuracy.Adversarial Machine Learning (AML): AML techniques have been used to test the robustness of DDoS detection models. By training the models using adversarial examples, the models are expected to be more robust against DDoS attacks.

It is important to note that DDoS attacks are constantly evolving, so AI-based DDoS detection techniques must also be updated and improved over time to remain effective. Additionally, the research in the field is active and new techniques and approaches are continuously being proposed.

## 4. Proposed Method

The proposed methodology introduces a machine learning agent as an intelligent approach to combat distributed attacks. This agent acts as a node that continuously monitors the network, generating alerts to all nodes. The agent can efficiently detect and prevent malicious traffic by making joint decisions and adjusting routing policies. The agent is composed of perception and cognitive modules, allowing it to take reactive and proactive actions. When malicious traffic is detected, a static agent controls the end host, and other agents transmit warnings to intermediate nodes about the malicious traffic that tracks middle network traffic. These agents can also track medium network traffic and use the bandwidth between nodes to make logs for labeling the TTL, ACL, and Packet. In case of any failure due to an attack or access block, these agents can follow alternate routes and alter the routing policies in routers accordingly.

Traditional security mechanisms against DDoS attacks include source-end, wrong end, and moderate IDS/IPS firewall systems. Researchers are inclined towards the third approach as it avoids casualties, but it has the drawback of requiring enormous resources to react quickly to detection. In the current scenario of DDoS attacks, attackers can quickly increase the power of their attack and secure the most available resources once they have breached the system. Researchers can also deploy attacks without spoofing IPs and apply simulations to complex exchanges of database queries, making it challenging to exclude such exchanges from the valid questions.

As shown in Figure 1, our proposed method consists of three main phases: training, feature selection and prediction, and traffic classification and detection.

In the training phase, we use PCAP datasets that contain network traffic data. We preprocess this data to extract relevant features, such as packet length, packet direction, and flow duration. We also use an open switch to simulate network traffic and capture more data for training.

Next, in the feature selection and prediction phase, we train different machine learning models on the preprocessed data. We select the best model based on its accuracy in predicting network traffic. Then, we use feature selection techniques to choose the most important features from the model to reduce computational complexity and improve the accuracy of our prediction.

Finally, in the traffic classification and detection phase, we use the selected features from the previous phase to classify the network traffic into different types, such as normal, malicious, or suspicious. We use the best feature and model combination to detect potential DDoS attacks and take appropriate action, such as blocking the source of the attack.

Overall, our approach leverages machine learning-based feature extraction and classification methods to detect DDoS attacks in real time with high accuracy and efficiency.

As shown in Figure 2, storing the labels of classified packets in a database is an essential step in our feature attack detection process. By doing so, we can analyze the patterns of malicious traffic and identify the characteristics of the attack. This database can be used for future reference, and we can continuously update it as new attacks occur.

Using the information stored in the database, we can create rules defining what constitutes an attack. These rules can be used to develop a feature-based detection model that automatically detects future attacks based on their features. The feature-based detection model is designed to identify specific patterns and features in the network traffic that are indicative of malicious behavior. These features can include source IP addresses, destination IP addresses, packet size, time of day, and many other variables.

We must select the most relevant features from the stored packet labels to create the best feature-based detection model. This process involves selecting the most discriminating features that can distinguish between malicious and benign traffic.

Once the most relevant features are selected, we can train a machine-learning model to detect these features in real-time. This model can be continuously updated using the latest attack data from the database.

Overall, storing packet labels in a database and using them to develop a feature-based detection model is an effective way to identify and prevent DDoS attacks. By leveraging machine learning and continuously updating our models, we can stay ahead of attackers and keep our network secure.

The described filter approach [31] is implemented in the Figure 3 flowchart and applied only at a single location, not multiple. For multi-location, we have extended it as follows. First, we define a flow as a septuple (source (IPv4/IPv6 address, transport port, and MAC address)) and destination (IPv4/IPv6 address, transport port, and MAC address). This method ensures that the same traffic flow traversing multiple data centers will be captured as individual flows in our data to analyze the attack traffic visibility at further different sites. Second, we define two variants of the detection threshold t. In the first variant (local threshold), only the detection is applied to traffic from a single ISP. In the second variant (global threshold), we detect a DDoS attack if the traffic sum exceeds t overall ISPs.

The detection flowchart shows the sequence of events necessary to formulate a sampling processing flow based on packet alarm. We propose an abnormal state based on a gathering approach working in a dispersed framework utilizing fitting fusion procedures to settle decisions. Conventional component extraction in a dispersed domain, including various sensors, and identifying DDoS attacks based on a class-particular subset of elements at the individual level, will give the base of the hidden layer of the proposed arrangement. Lastly, personal decisions can be joined utilizing a suitable blend governed at the upper layer. Many autonomous agents in our proposed architecture aim to secure the internal system network. The agents would be sent warning signals to a network administrator in any possible attack-related scenarios. Each agent would then perform a security monitoring function at a specified node and track the network traffic data toward any attack. The agents would then send warning messages relating to the attack to its server. There is an authentication position between the victim and the server to validate the hop-count information used by the filtering process for hop-counting. Every agent provides cost-effective control of network traffic at a low flow level. The recognition method is proposed to detect sudden changes in network traffic flows at the level of the agents’ antonyms. The DDoS attacks initially target the deployed agents, and they read variations within the network in the spatial temporal allocation of network traffic volumes. Such variations in network traffic flows are typically directed at the victim node, although random variations due to natural traffic flows are not in the victim’s direction. Based on these improvements, these agents exchange details about traffic surges with each other and collaboratively recognize the attacker. A warning message to the source node is then sent upstream. The agents perform the detection of DDoS attacks, which were initially deployed close to the source. The cycle then continues from agent to agent, continuously against the victim. Different scenarios, such as regular and DDoS attacks and normal and flash, compare the source address entropy and traffic cluster entropy.

The following explanations employ simple words and symbols. The packet’s source IP address is a 4-byte logical address (S_IP). It is called a traffic cluster (TC) when all the traffic comes from a single set of networks or administrative domains. All the packets with the same first 16 bits belong to the same group, referred to as the 16-bit traffic cluster. Bit-wise AND a 16-bit mask return 255.255.0.0 as a result. A 16-bit traffic cluster (16_TC) identification is the name given to the unique identifier assigned to a traffic group or cluster of this type. If all packets in a segment share the same first 24 bits, they are considered part of the same 24-bit traffic cluster. Bit-wise AND a 24-bit mask give 255.255.255.0 as a result, which is then converted to hexadecimal. A 24-bit traffic cluster (24_TC) identification is the name given to the unique identifier assigned to a traffic group or cluster of this type.

## 5. Results and Analysis

The detection system uses different sampling rates in a controlled network environment to classify network traffic. To conduct this experiment, we used raw network traffic from the CICIDS2017 and CSE-CIC-IDS2018 datasets and traffic from a local testbed. A total of five virtual machines were used in the experiment, each with four virtual CPUs (vCPUs) and 4 GB of RAM. The intelligent detection system attained low false alarm rates. 

### 5.1. Description of the Benchmark Datasets

Researchers recommend using various data sets, such as the DARPA (Lincoln Laboratory 1998–99), KDD’99 (the University of California Irvine 1998–99) [6], and LBNL (Lawrence Berkeley National Laboratory and ICSI 2004–2005) to analyze the effectiveness of their detection and prevention strategies. The problem is that the information in these datasets is stale. HOIC locally created datasets and LOIC datasets contained the most recent threats and DoS technologies for use in this research.

#### 5.1.1. The CIC-DoS Dataset

In combination with the ISCXIDS2012 dataset without an attack track, the CIC-DoS dataset concentrates on the application layer DoS attacks. Eight DoS attacks were generated at the application layer [32]. The resulting data set contains 4.6 GB of 24-h network traffic and is freely available on the Internet (https://www.unb.ca/cic/datasets/dos-dataset.html (accessed on 20 January 2021)). The CIC-DoS attack incidents and tools are summarized in Table 2 and Table 3 with different datasets. When using the program slow httptest to generate high-low-volume attacks, the default value of the attack traffic is 50 connections per attack. This makes attacks more suspicious, as per [32].

#### 5.1.2. The CICIDS2017 Dataset

ISCX created the CICIDS2017 data set in 2017, including both legitimate and malicious attack traffic. For the first time, the IDS data set comprises seven typical update attack families that meet the basic standards and are available to the public (http://unb.ca.cic.datasets/IDS2017/ (accessed on 15 August 2021)). The primary emphasis of this research is the malicious DoS activity recorded on Wednesday, 5 July 2017, by a file including 5 DoS/DDoS attacks and diverse network traffic. The data set includes 13G worth of network activity over eight hours. The attack tools utilized are Holk, low loris, heartbleed, Slow httptest, and GoldenEye.

#### 5.1.3. CSE-CIC-IDS2018 Dataset

The Communications Security Agency (CSE) and the Canadian Cyber Security Institute (CCSI) collaborated on the development of this data set (CIC) are available to the public (https://www.unb.ca/cic/datasets/ids-2018.html (accessed on 10 December 2021)). A total of seven assault scenarios are included in the final data set: Heartbleed and Brute Force, as well as Botnets, Web Attacks, DoS attacks, and DDoS attacks. There are 250 machines in the attack infrastructure and 620 laptops and desktops, and 40 servers in the victim organization’s network. Dataset analysis and other related principles have been described in detail in published research materials [33,34].

This effort focused on finding malicious DoS/DDoS activity in files. SlowHTTPTest, Slow httptest, Hulk, LOIC, and HOIC are the attack tools utilized.

#### 5.1.4. CIC-DDoS2019

The CIC-DDoS2019 dataset was used to train a Long Short-Term Memory (LSTM) and Convolutional Neural Network (CNN) model to detect Distributed Denial of Service (DDoS) attacks and the dataset are available to the public (https://www.unb.ca/cic/datasets/ddos-2019.html (accessed on 18 January 2022)). The first step in this process was to prepare the data, including pre-processing and normalizing it. The pre-processed data was then split into training and testing datasets. The LSTM and CNN models were trained on the training data and then evaluated on the testing data. Finally, the performance of both models was also compared in terms of accuracy, precision, recall, and other performance metrics. The better-performing model was selected and further fine-tuned for deployment in a real-world DDoS detection system and is pending for future work.

#### 5.1.5. Local Generated Dataset by Testbed

The customized dataset was developed in a controlled network environment.

As shown in Figure 4 we created VLANs 1, 49, 59, 79, 89, 99, and 40 for the targeted host as a victim of the attacker. VLAN 165 is only applicable to users of training units. VLAN 69 is used as the attack host, and monitoring is performed on VLAN 1. All networks can access the Internet normally. The attack plan was configured to generate an attack every 30 min from 00 h 00 m 00 s to 23 h 59 m 00 s within 24 h. All attacks were executed by the attacker’s host 192.168.60.100, during which no legitimate traffic was sent to the victim. The attack tool was parameterized and can generate covert small, medium, and lightweight modes and large-scale attacks. Table 4 shows the eight types of protocol-based and application-based attacks that were used for four different attack tools. The duration of protocol-based attacks and application-based mass attacks is 30 s, while the period of application-based small-batch attacks is 30 to 240 s.

When we used the slow HTTP test tool to complete a minor attack, the number of connection parameters was 3000 instead of the default option of 100 connections.

### 5.2. Lab Experiments

We conducted lab experiments in the IT lab in four steps. The first step was to examine the traffic data from the traffic captured files (.pcap) and collect 90 days of traffic. In the second step, we designed an attack strategy and linked and analyzed different parameters from collected data, such as the source addresses, destination addresses, attack type, and attack duration. In step three, we classified the traffic and performed preprocessing. After classification, in step four for evaluation, we compared the attack plan with results to ensure that the system performed as expected.

We also included traffic data from the most used datasets, such as CIC-DoS, CICIDS2017, and CSE-CICIDS2018, with custom-generated data sets to meet the first two steps of the verification process. Traffic preprocessing and classification were set up as detailed in step three. We used TcpReplay software and an intelligent detection system with two virtual Linux machines using OVS (Open Virtual Switch) in the lab experiments (see Figure 5).

### 5.3. Result Discussion

The proposed approach was evaluated using the above datasets, system setup, and metrics. One of the most common findings was that the CSE-CIC-IDS2018 data set yielded the most precise results when all three parameters were set to 100%: 100 percent.

“DR” refers to “detection rate”, which measures the proportion of actual positive cases that were correctly identified as positive by the algorithm or model being evaluated. An 80% detection rate meant that the algorithm correctly identified 80% of the positive cases. “FAR” stands for “false alarm rate”, which measures the proportion of negative cases that are incorrectly identified as positive. A 2% false alarm rate means that only 2% of the negative cases were falsely identified as positive. “PREC” refers to “precision”, which measures the proportion of true positive cases among all cases that were identified as positive. A precision of 99.20% means that 99.20% of the cases that were identified as positive were actually true positives.

In [34], the performance of different support vector machine (SVM) kernels in an intrusion detection system (IDS) using the Principal Component Analysis feature selection technique was evaluated. The paper has a clear methodology but is limited by the use of only two datasets and a lack of comprehensive discussion of related work and detailed analysis of results. Based on this information, it appears that the CICIDS2017 dataset had the lowest performance compared to the other datasets that were evaluated; while it had high precision (99.20%), its detection rate was only 80% and its false alarm rate was 2%.

#### Performance Measures

The performance measures from [35] are used here and described below.
(1)$$\mathrm{Specificity}\;=\frac{TN}{FP+TN}$$
(2)$$F1\;\mathrm{score}\;=\frac{2TP}{2TP+FP+FN}$$
(3)$$\mathrm{Accuracy}\;=\frac{TP+TN}{TP+TN+FP+FN}$$
(4)$$\mathrm{Precision}\;=\frac{TP}{TP+FP}$$

Figure 6 and Figure 7, Table 5 and Table 6 summarize the system performance for each dataset and other machine learning algorithms.

Despite multiple failures in this more realistic scenario, the proposed system performed well enough to be competitive. On the other hand, the CICIDS2017 dataset produced the poorest results, with a DR of 80%, a FAR of 2%, and a PREC of 99.20%. This set of data represents a more realistic network scenario, with a mix of legitimate traffic and a significant proportion of malicious traffic (such as slow application-layer attacks). This method can be implemented using the proposed system which detects four-fifths of attacks with a PREC of more than 1% and greater than 90%. We will use the CICIDS2017 dataset to talk about online detection and how much computing power was used throughout the experiment. This dataset summarizes the primary DoS attack routes as a natural and recent phenomenon. OpenSSL’s vulnerability, described in CVE-2014-0160, is used in this attack to collect data, but it is expected to behave like a DDoS in any application. As a result, the system returns false negatives. Heartbleed attacks without a denial-of-service attack or statistical matching in traffic samples are the most evident causes of this FN and FNR. The first attack takes advantage of an excellent regular connection, whereas the second uses a sample that matches a legitimate traffic signature. The results of this study using the 2017 dataset are compared with a model presented in [35] having identical performance and are reported in Table 7. Thus, we conclude that all our results are comparable with the existing literature.

### 5.4. False Positive Rates of DDoS Detection Models

False positives occur when a DDoS detection model incorrectly identifies benign network traffic as malicious, leading to unnecessary alerts and potential disruption of legitimate traffic. Minimizing the false positive rate is crucial in maintaining the performance and usability of a detection system. In this graph, we compare the false positive rates of our proposed model with those of two popular models—Convolutional Neural Network (CNN) and K-Nearest Neighbors (KNN)—using different datasets.

Our model outperforms the other two models in terms of false positive rate, indicating its superior ability to accurately distinguish between benign and malicious traffic as shown in Figure 8.

The experiment was typically completed in a live environment, and the DDoS attack live detection is shown in Figure 9. The overall network traffic demonstrated in Figure 9 highlights the DDoS attack load sent to the detection system, and Figure 10 presents an analysis of the pcap traffic. As can be seen, for network traffic with the highest delay, the detection system receives normal and malicious traffic, making this approach scalable. Our results show that the system efficiently distinguished legitimate traffic from DDoS attacks.

## 6. Conclusions

This paper studies the detection of Distributed Denial of Service (DDoS) attacks using machine learning algorithms and aims to develop an intelligent detection agent solution to prevent DDoS attacks. The proposed solution uses an efficient adaptive feature selection approach to increase detection rates up to 99.8%, reduce false alarm rates, and minimize computation and transmission costs. The machine learning agents are trained to watch network traffic and analyze packets to identify potential attacks. The proposed approach focuses on the defense before the attack, as prevention is better than cure. The study covers all attack domains, tools, and countermeasures. The results of our study show that the proposed solution has the potential to provide significant protection against DDoS attacks. However, we also acknowledge that there is still room for improvement and further research. This paper serves as a foundation for multi-directional research on the topic and provides a comprehensive study on the use of three popular machine learning algorithms to detect DDoS attacks while highlighting the importance of a proactive defense mechanism. While the proposed solution may not be completely neutralizing, it provides a significant step towards effectively defending against DDoS attacks.

## Figures and Tables

**Figure 1 sensors-23-03333-f001:**
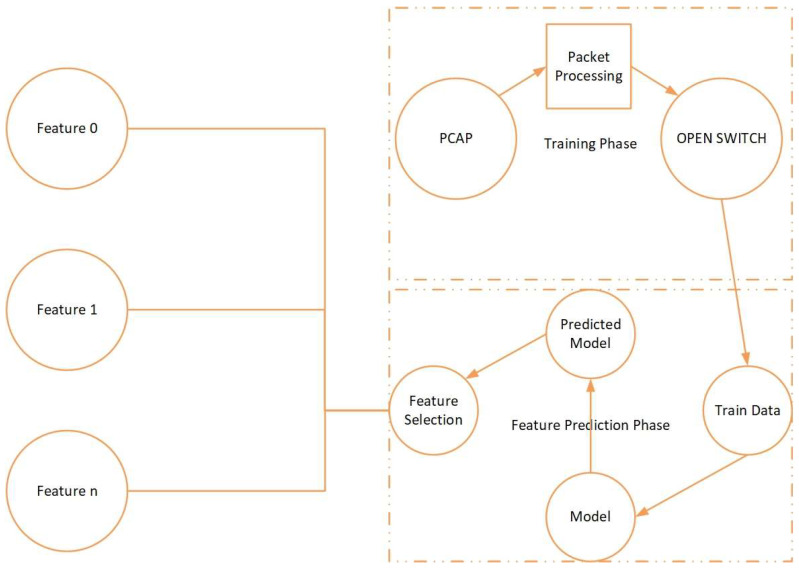
Proposed detection method.

**Figure 2 sensors-23-03333-f002:**
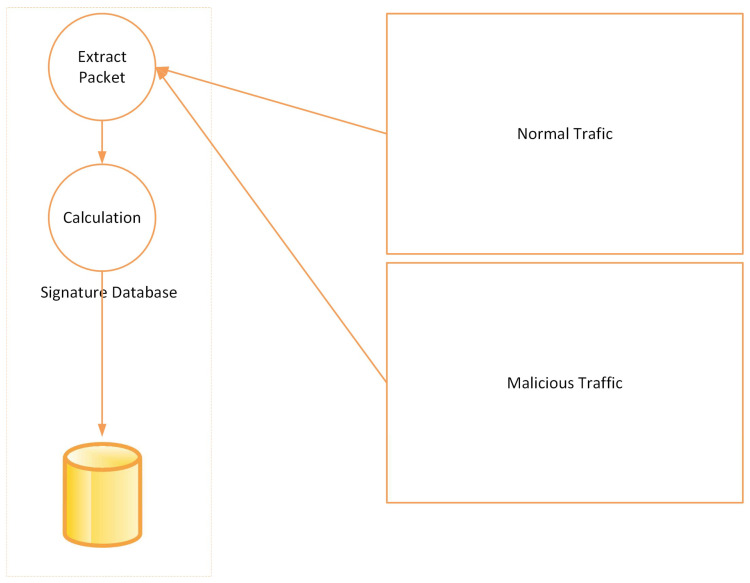
Storing the labels of classified packets in a database.

**Figure 3 sensors-23-03333-f003:**
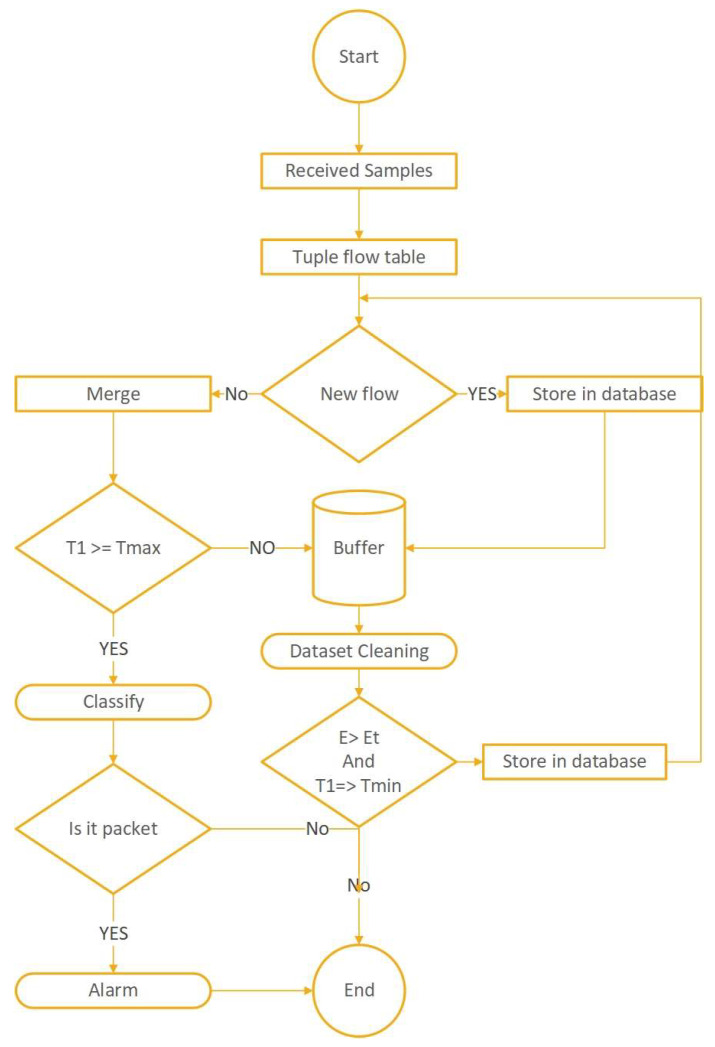
Proposed detection system flowchart.

**Figure 4 sensors-23-03333-f004:**
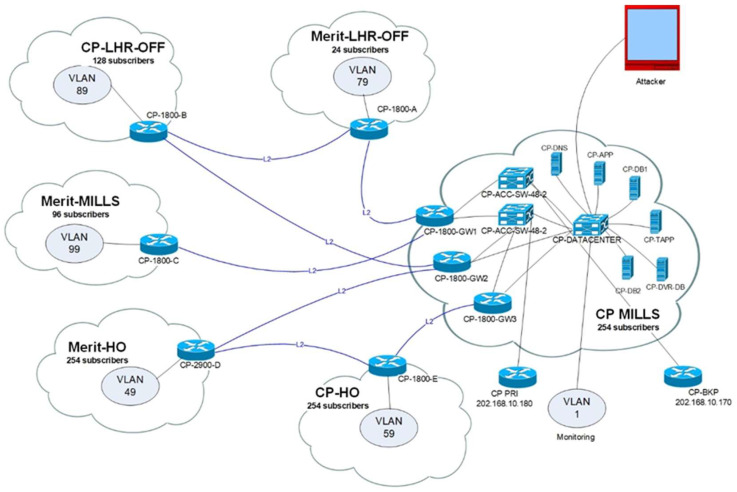
Local testbed design.

**Figure 5 sensors-23-03333-f005:**
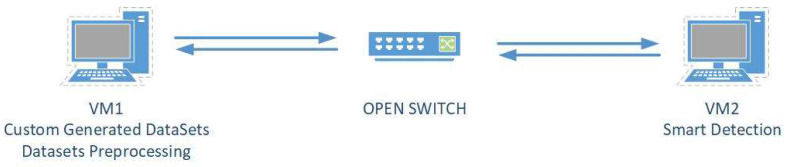
Detection validation system.

**Figure 6 sensors-23-03333-f006:**
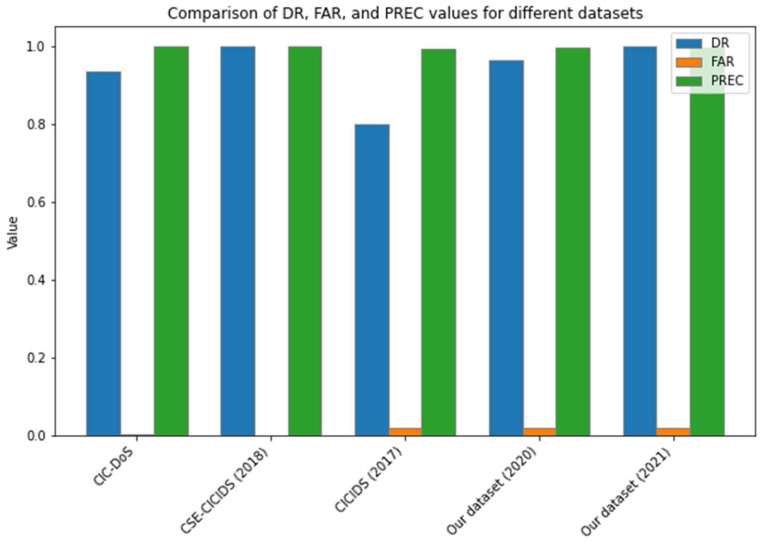
System performance graph.

**Figure 7 sensors-23-03333-f007:**
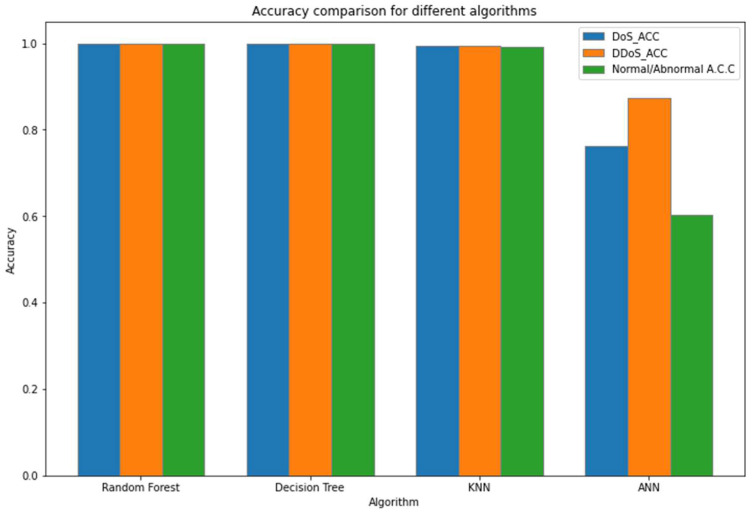
Accuracy comparison for different algorithms.

**Figure 8 sensors-23-03333-f008:**
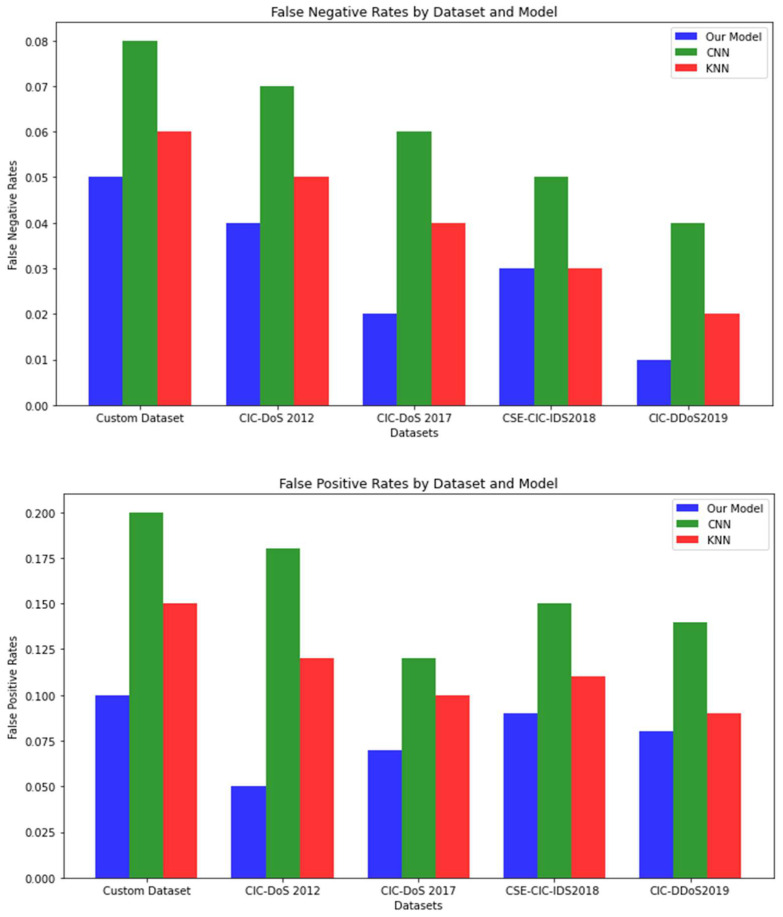
False positive and false negative.

**Figure 9 sensors-23-03333-f009:**
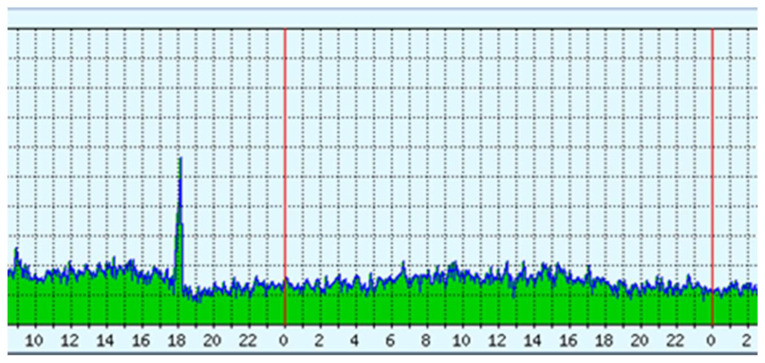
DDoS traffic live detection.

**Figure 10 sensors-23-03333-f010:**
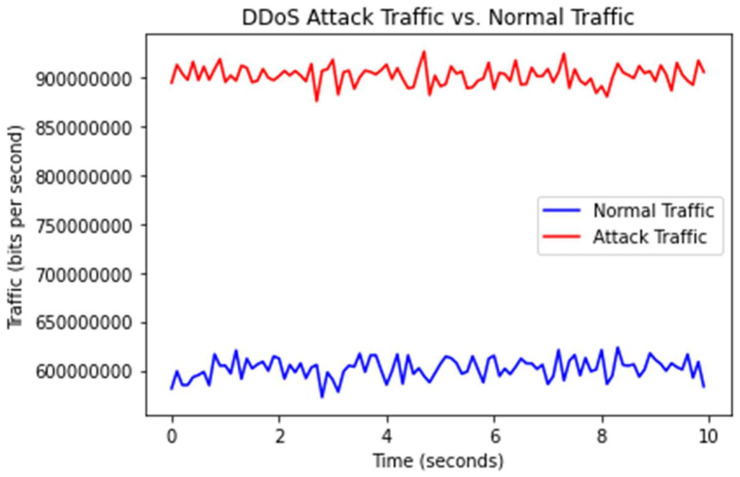
DDoS traffic and normal traffic.

**Table 1 sensors-23-03333-t001:** Comparison of the literature.

Reference Number	Method	Drawback	ACC
[16]	Combined EA, SVM, and ANN	Limited dataset	99.3%
[18]	Hybrid-based IDS	Fixed set of features	96.64%
[19]	ML IDS for MIoT	Single dataset used	99.9%
[21]	Dynamic Anomaly Detection Scheme	Only on AODV-based	84.0%
[24]	Mix machine learning techniques	Small dataset, fixed features	99.5%
[25]	Combined DTF, CNN, and LSTM	Only wormhole detection	96%
[26]	Web-based DDoS detection	Only web single dataset	99%
[27]	Mining sequences of IP’s	Some worst performance	-
[28]	Building and evaluation using ANN-MLP	Single dataset: UNSW-NB15	76.96%
[29]	Detection by ensemble of neural classifiers	Overfitting	99.4%
[30]	Detection by MLP, NB, and RF	Not applicable for all attacks	98.63
[35]	Detection by CNN and LSTM	Not applicable for low volumes	96.7

**Table 2 sensors-23-03333-t002:** CIC-DoS 2012 dataset.

No	Description of Dataset	Total Number of Events
1	ddossim	2
2	Golden Eye	3
3	DoS GET	4
4	slowbody2	4
5	slowread	2
6	slowheaders	5
7	rudy	4
8	slowloris	2

**Table 3 sensors-23-03333-t003:** CIC-DoS 2017 Dataset.

No	Description of Dataset	Total Number of Events
1	DDoS	128,027
2	Golden Eye DoS	10,293
3	BENIGN	2,273,097
4	Hulk Dos	231,073
5	PortScan	158,930
6	Slowhttptest Dos	5499
7	Hearbleed	11
8	Slowloris DoS	5796

**Table 4 sensors-23-03333-t004:** CIC-DoS 2017 dataset.

Events	Testbed Attack Types
4	TCP SYN flood
4	TCP SYN flood-light mode
4	TCP ACK flood
4	UDP flood-random DST port
5	DoS improved GET
4	DoS GET
5	Slow body
4	Range attack

**Table 5 sensors-23-03333-t005:** System performance evaluation with datasets.

Datasets	DR	FAR	PREC
CIC-DoS	0.936	0.04	0.999
CSE-CICIDS (2018)	1.000	0.00	1.000
CICIDS (2017)	0.800	0.02	0.992
Custom dataset Collected (2020)	0.965	0.02	0.995
Custom dataset Collected (2021)	0.998	0.02	0.995

**Table 6 sensors-23-03333-t006:** Algorithm performances.

Algorithm	DoS_ACC	DDoS_ACC	Normal/Abnormal A.C.C.
Random Forest	0.999272	0.999977	0.999511
Decision Tree	0.999567	0.999935	0.999559
KNN	0.994917	0.995384	0.991968
ANN	0.763600	0.873800	0.603400

**Table 7 sensors-23-03333-t007:** Accuracy compared with CNN-based model.

Paper	ML Model	A.C.C.	F1	Pr
[35]	CNN	96.7	98.18	97.59
Our	KNN	99.87	99.87	99.84

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
